# Block and Subword-Scaling Floating-Point (BSFP) : An Efficient Non-Uniform Quantization For Low Precision Inference

**Yun-Chen Lo, Tse-Kuang Lee, Ren-Shuo Liu**
Department of Electrical Engineering, National Tsing Hua University, Hsinchu, Taiwan

## Abstract

In this paper, we propose **B**lock and **S**ubword-Scaling **F**loating-**P**oint (**BSFP**), a datatype with a non-uniform quantization scheme for the skewed and non-uniform distribution of weight vectors in neural networks. By quantizing each weight vector as the superposition of multiple subword vectors (in two's complement) with scaling factors (in Low-bit Floating-Point, LBFP), BSFP can effectively fit the distribution of weight vectors while maintaining high computation efficiency. Furthermore, we present a grid search-based MSE-optimal quantization flow and a scaled serial processing engine to complete the quantization pipeline and the infrastructure.

The experimental results on the ImageNet classification task show that our proposed method outperforms state-of-the-art Microsoft Floating Point (MSFP) by up to 18.57% top-1 accuracy at the same weight precision and reduces up to 10.3% model size. Furthermore, BSFP outperforms MSFP by up to $2.0\times$ computing throughput and up to $5.3\times$ energy efficiency under the same silicon area budget.

## 1 Introduction

Deep Neural Networks (DNNs) have continuously enabled more and more eye-catching artificial intelligence (AI) applications Johnson et al. (2016); Lin et al. (2014); Deng et al. (2009). However, their large model size and high computational complexity hinder the wide deployment of DNNs to latency-sensitive cloud services and energy-constrained edge devices. To address the performance and energy challenges, in addition to compacting neural network structures Sandler et al. (2018); Ma et al. (2018), reducing the bitwidths of weights or activations also have been extensively explored Jacob et al. (2018); Darvish Rouhani et al. (2020); Tambe et al. (2020); Li et al. (2020).

Particularly, non-conventional datatypes and custom hardware are emerging to optimize the performance, energy efficiency, area efficiency, and memory requirements of DNN inference. Prior industry and academia researches have explored low-bit floating-point datatypes Kalamkar et al. (2019); Jouppi et al. (2020); NVIDIA (2022); Tambe et al. (2020), block-based floating-point datatypes Darvish Rouhani et al. (2020); Köster et al. (2017), low-bit fixed-point datatypes NVIDIA (2020); Jacob et al. (2018), and power-of-two fixed-point datatypes Miyashita et al. (2016); Zhou et al. (2017); Li et al. (2020) as the potential candidates in efficient DNN inference. Among many datatypes, Microsoft Floating Point (MSFP), a kind of block-based floating-point type as shown in Figure 1(b), claims to achieve the state-of-the-art tradeoff among dynamic range, DNN accuracy, and hardware complexity Darvish Rouhani et al. (2020).

This work focuses on post-training quantization, which is preferable in practice. First, for end users, it involves no data (including private data) and enables a low-friction deployment pipeline Nagel et al. (2019). Second, according to our discussions with an IC design house that tapes out AI chips in advanced technology nodes, the industry (at least their application-side customers) does appreciate post-training quantization because, in most cases, AI application companies are reluctant to release AI models and training data to AI accelerator companies. Although we focus on post-training quantization, we still include the fine-tuning results in Appendix A.

This paper proposes **Block and Subword-Scaling Floating-Point (BSFP)**, a new class of datatypes with a bit-efficient, non-uniform quantization method and custom hardware to improve the energy

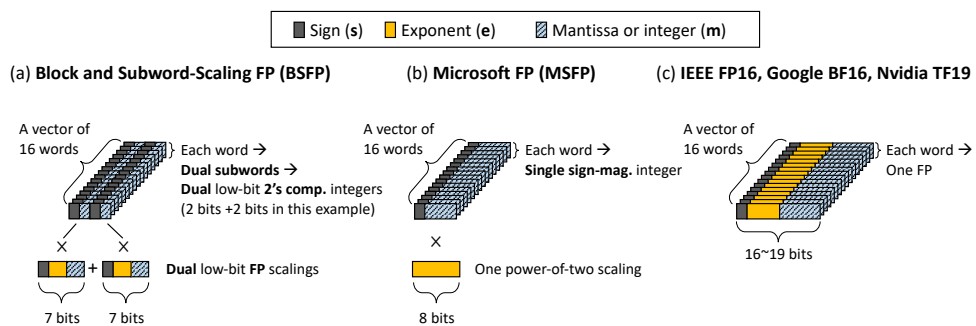

Figure 1: Number system comparison between (a) the proposed Block and Subword-Scaling Floating-Point (BSFP), (b) Microsoft FP (MSFP Darvish Rouhani et al. (2020)), and (c) floating-point numbers (IEEE 754 FP16, Google BF16 Jouppi et al. (2020), and Nvidia TensorFloat (TF19) NVIDIA (2022)).

efficiency and performance over state-of-the-art MSFP. As shown in Figure 1(a), the key idea of BSFP is to *approximate each full-precision weight vector using the sum of two subword vectors with two scalings, respectively. More specifically, each subword is a low-bit (e.g., 2-bit), signed (two's complement) integer, and each scaling is a low-bit floating-point (LBFP) number (e.g., a 7-bit one).* We will show that BSFP is superior to MSFP in capturing the nonuniformity and skewness of per-vector weight distributions, which are common cases for a vector of a small number (e.g., 16) of weights. In addition, although BSFP adopts two scalings and two subword vectors, it can still be efficiently computed for the following three reasons. First, the computation cost of scaling is amortized over 16 weights. Second, each scaling is an LBFP and involves only low-bit operations, e.g., multiplications with a 3-bit mantissa. Third, the subword vector structure happens to fit bit-serial computation architectures Qian Zhang et al. (2022); Judd et al. (2016).

One property that BSFP exhibits is to approximate the desired weight vector using both coarse and fine vectors. One subword vector with a large scaling captures large weights, and the other subword vector with a small scaling mitigates the remaining deviations. Therefore, BSFP can adapt to large outliers and small resolutions simultaneously.

Figure 2(a) compares the quantization results of a real 16-element weight vector from ShuffleNet-v2 in either 8-level BSFP or 15-level MSFP. This example clearly demonstrates the potential that even BSFP with relatively fewer quantization levels can achieve smaller quantization errors (e.g., in terms of MSE) than MSFP with more quantization levels. We summarize the rationales for BSFP's superiority below:

- **No waste of quantization level**: BSFP utilizes two's complement for each subword and does not waste precious quantization levels. In comparison, MSFP resembles sign-magnitude and wastes one quantization level (i.e., duplicated $+0$ and $-0$). Even worse, the impact of wasting quantization levels increases as the bitwidth goes down. For instance, a 3-b two's complement number can represent eight quantization levels, 12.5% more than the seven levels of the 3-b sign-magnitude number.

- **Adaptation to skewed distribution**: BSFP exploits the asymmetrical nature of two's complement numbers (e.g., -2, -1, 0, 1 for 2-b two's complement numbers) and the sign of the associated scaling to adapt to the asymmetrical weight distribution of in weight vectors. In comparison, MSFP is permanently restricted to symmetrical quantization levels and leads to a waste of quantization levels fitting asymmetrical distributions.

- **Adaptation to non-uniform distribution**: BSFP can offer non-uniform quantization levels by combining two subword-scaling vectors. In comparison, MSFP always uniformly quantizes weight vectors, which may instead exhibit non-uniform weight distributions.

- **Better freedom of quantization step size**: The quantization step size of BSFP is defined by the two scalings, which are (low-bitwidth) floating-point values. In contrast, the quantization step size of MSFP cannot be any value other than power-of-two, e.g., 0.5, 0.25, 0.125.

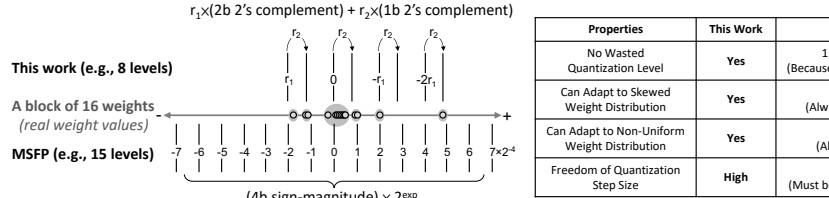

Figure 2: (a) Quantizing **16 real weights of ShuffleNet-v2** using 3-b Block and Subword-Scaling Floating-Point (BSFP) can achieve both lower quantization error (MSE) and lower storage than 4-b Microsoft Floating Point (MSFP). (b) Properties comparison of BSFP with MSFP.

BSFP can be deployed to servers and edge devices, achieving high performance and energy efficiency. Please note that the main advantages of BSFP are not from specific MAC architecture. Therefore, we select one MAC architecture and focus on comparing BSFP to MSFP datatype on that architecture. This is also the case for the MSFP paper, which compares MSFP against BF16 and INT). Our experiments will demonstrate the robustness and generality of BSFP using several mainstream DNNs on the ImageNet classification task. In summary, this work makes the following contributions:

- We propose Block and Subword-Scaling Floating-Point (BSFP), a hardware-algorithm co-designed numerical datatype for DNNs that can achieve higher accuracy with better computing throughput and energy efficiency than MSFP.
- We design a scaled serial processing engine to support the proposed BSFP. This custom hardware enjoys a small area footprint and can support various configurations of BSFP.
- We identify mean squared error (MSE) as an effective criterion in determining the LBFP scaling factors of BSFP and a grid search-based MSE-optimal quantization flow to enable a low-friction deployment pipeline. We present both post-training and fine-tuning procedures for preparing DNNs in the proposed BSFP format.
- We perform extensive evaluations on various DNNs of the ImageNet classification task and demonstrate that BSFP successfully outperforms MSFP in model size, quantization error, accuracy, throughput, and energy efficiency.

## 2 RELATED WORKS

There is rising attention to designing custom datatypes for efficient inferencing. The wide range of formats can be categorized into four classes, i.e., low-bit floating-point, low-bit fixed-point, power-of-two, and block-based floating-point.

The first category, low-bit floating-point numbers (FP16, Bfloat16 (BF16), and TensorFloat), simplifies the IEEE-754 floating-point formats. Representative commercial hardware include Google's TPUs (BF16) Jouppi et al. (2020) and NVIDIA's A100 GPUs (Tensorfloat, TF19) NVIDIA (2022).

The second class of datatype is low-bit fixed-point datatype (e.g., INT4), whose operations are equivalent to integer operations and can achieve low hardware cost and high performance. Such fixpoint numbers include 8-b Jacob et al. (2018), 4-b NVIDIA (2020); Dai et al. (2021), 3-b Mellempudi et al. (2017), and ultimately binary Hubara et al. (2016). Although fixed-point datatype achieves a small hardware area and receives great popularity, it requires careful model re-calibration and suffers from a large accuracy drop when representing values with a high dynamic range.

The third class is power-of-two numbers, including Power-of-Two Zhou et al. (2017); Miyashita et al. (2016) and Additive Power-of-Two Li et al. (2020), which utilizes one or two power-of-two terms to approximate floating-point weights. The power-of-two format is appealing for replacing multipliers with low-cost shifters. However, the available values are limited by the power-of-two form.

Additionally, some prior research also explores non-uniform quantization. For example, LQ-Nets Zhang et al. (2018) learns quantization levels to minimize the quantization error. Distillation Polino et al. (2018) optimizes the quantization levels by learning to minimize a task loss with their teacher networks. However, these methods use a limited number of floating-point numbers

to quantize the full-precision data points, leading to significantly larger overhead than fixed-point computation.

The last category is block-based floating-point, which forces a block of (e.g., 16) floating-point numbers to share one exponent. The block-based floating-point offers low area overheads and a large dynamic range. This format has enabled state-of-the-art accuracy-to-area Pareto frontier. Representative examples include Intel's Lake Crest (Flexpoint Köster et al. (2017)) and Microsoft's Brainwave Fowers et al. (2018) (MSFP Darvish Rouhani et al. (2020)).

The closest work related to this paper is MSFP in NeurIPS 2020 Darvish Rouhani et al. (2020), which is the current state-of-the-art format in terms of area-to-accuracy trade-offs. Our BSFP design is fundamentally different and novel compared to MSFP for the following reasons: 1) BSFP approximates full-precision weight vector using the superposition of multiple subword-scaling vectors, which MSFP and other prior works do not explore before. Please refer to the properties comparison in Figure 2 and the corresponding discussion of BSFP's superiority. 2) We additionally design a bit-serial processing engine that can support different configurations of BSFP. In comparison, the MSFP work considers a bit-parallel processing engine and does not support format changes.

## 3 BLOCK AND SUBWORD-SCALING FLOATING-POINT (BSFP)

We propose BSFP that quantizes an $l$-element full-precision weight vector ($\overrightarrow{W_{fp}}$) using $N_{sub}$ $l$-element subword-scaling vectors ($\overrightarrow{W_{sub_i}} \times scale_i = \overrightarrow{W_{sub_i}} \times (-1)^{s_i} 2^{e_i} m_i$), where the subwords adopt the two's complement format and the scaling factors adopt the LBFP format. The default $l$ is 16 unless explicitly mentioned. These scaling factors $(-1)^{s_i} 2^{e_i} m_i$ determine the quantization points that a BSFP vector can represent. Since using two subwords achieves good accuracy-storage trade-offs, we set $N_{sub} = 2$.

$$\overrightarrow{W_{fp}} \approx \sum_{i=1}^{N_{sub}} \overrightarrow{W_{sub_i}} \times (-1)^{s_i} 2^{e_i} m_i \xrightarrow{N_{sub}=2} (\overrightarrow{W_{sub_1}} \times (-1)^{s_1} 2^{e_1} m_1) + (\overrightarrow{W_{sub_2}} \times (-1)^{s_2} 2^{e_2} m_2)$$

We apply BSFP to weights. In comparison, activations are left as MSFP as the MSFP paper does Darvish Rouhani et al. (2020). The rationales behind this decision are as follows. Weights differ from activations in that weights are available offline. Thus, weights can and also should enjoy a longer quantization time budget. Clearly, if both weights and activations adopt MSFP, as the MSFP paper does, it directly suggests that some optimization opportunities for weights are left on the table.

**Computing with BSFP format.** Dot products are the fundamental operations in DNNs. Here we show the dot product of an $l$-element $\overrightarrow{W_{bsfp}}$ weight vector and a $\overrightarrow{A_{msfp}}$ activation vector.

$$\overrightarrow{W_{bsfp}} \cdot \overrightarrow{A_{msfp}} = (\sum_{i=1}^{N_{sub}} \overrightarrow{W_{sub_i}} \times (-1)^{s_i} 2^{e_i} m_i) \cdot (\overrightarrow{A_{msfp}} \times 2^{e_{shared}})$$

$$= \sum_{i=1}^{N_{sub}} ((\overrightarrow{W_{sub_i}} \cdot \overrightarrow{A_{msfp}}) \times ((-1)^{s_i} m_i 2^{e_i + e_{shared}}))$$

For each subword vector of BSFP, it first multiplies with the MSFP activation vector using compact multipliers and an adder tree to obtain the partial sum. Secondly, the LBFP scaling factor of BSFP is combined with the shared exponent of MSFP to get the correct scaling factor. More specifically, the exponent fields of BSFP and MSFP are summed together. Finally, we scale the partial sum with the combined scaling factor. The hardware overheads of BSFP over MSFP is to multiply the partial sum by a low-bit (3-b or 4-b) mantissa of the scaling factor.

**Scaled Serial Processing Engine ($S^2PE$).** Figure 3 shows the proposed scaled serial processing engine ($S^2PE$) and its example in computing a 4b BSFP weight vector. Within an $S^2PE$, the 2b multipliers are chosen to enable flexibility in computing different bitwidth configurations with bit-serial computation Qian Zhang et al. (2022). This design choice achieves a compact silicon footprint and high computation flexibility.

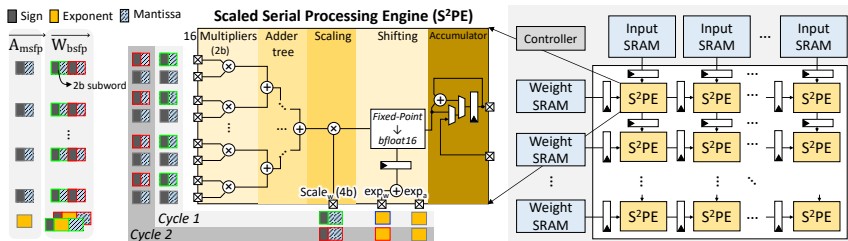

Figure 3: Proposed scaled serial processing engine (S²PE) and systolic architecture.

In addition, the proposed PE only requires an integer multiplier to scale the partial sum by the (at most) 4-b mantissa of the LBFP scaling factor. The exponent terms of BSFP and MSFP are added together to convert the scaled partial sum back to BF16 for accumulation. For BSFP with a bitwidth larger than 2, the proposed PE computes the result using multiple cycles. We finally present a systolic architecture, which integrates multiple scaled serial processing engines.

**Criterion-optimal quantization flow.** The flexibility of BSFP requires a flow to determine a suitable LBFP scaling factors setup. We observe that the number of combinations for LBFP scaling factors is limited; hence exhaustively evaluating all of them on GPU is feasible. Specifically, every weight vectors are parallelly quantized and evaluated using the same LBFP scaling setup. For example, an optimal setup for ShuffleNet-v2 can be found within 30 minutes using one NVIDIA V100 (ResNet-50: less than 1 hour; ViT: less than 1.5 hours).

Figure 4 shows the quantization flow, which consists of four steps: 1) *Generating subword configurations*, which generate several potential subword configurations that satisfy the target weight bitwidth. 2) *Generating LBFP scaling combinations*, which creates search space by exploring all potential combinations (grids) of LBFP scaling factors. 3) *Iterative rounding*, which quantizes the full-precision weight vectors using generated scaling factor combinations. 4) *Criterion evaluation*, which evaluates the difference between the full-precision weights and quantized counterparts using chosen criterion. The criterion candidates are L1, Mean Square Error (MSE), and Cosine Similarity, where MSE is selected as the final criterion. The details of the flow are presented in Appendix D.

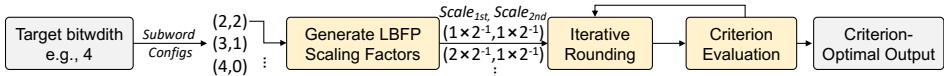

Figure 4: Criterion-optimal quantization flow for BSFP format.

# 4   BSFP CONFIGURATIONS

This section discusses the effects of different configuration setups for BSFP.

**Block Size (or vector length, $l$).**   The vector length affects the model size, the PE area efficiency, and the model accuracy. A large vector length generally amortizes the hardware overheads and reduces the storage overheads of LBFP's scaling factors. In practice, we found vector lengths of 16 to 64 to be effective for BSFP in preserving accuracy while incurring a moderate hardware cost.

**Quantization Criterion.**   The quantization criterion plays an important role in our quantization flow to decide the final accuracy. Common criterion candidates are Manhattan distance (L1 distance), Mean square error (MSE, Euclidean distance, L2 distance) Zhao et al. (2019), and Cosine Similarity Zhu et al. (2019); Zhang et al. (2019). We identify MSE as a better criterion in our quantization flow. Furthermore, users can easily define their quantization criterion in the framework.

**Number of Subwords.**   We can tune the number of subwords ($N_{sub}$) factor to explore the trade-offs for BSFP datatype. To fully obtain the benefit of BSFP, we suggest setting $N_{sub}$ larger than 1, e.g., 2 to 4. In practice, the number of subwords is set to 2 to balance storage overheads and accuracy.

**Low-bit Floating-point (LBFP) Scaling Factor Format.** The configuration for LBFP scaling factor format is also crucial in determining our quantization flow's accuracy and search space. Appendix H shows that scaling format with 1s4m3e and 1s3m3e achieves good trade-offs.

Besides, we propose to equip separate exponent biases for different LBFP scaling factors. The key idea is to set suitable biases to minimize the LBFP scaling factor's precision while achieving satisfying accuracy. We empirically set exponent biases of the first and second subword scaling factors to be -3 and -8, respectively. It is noteworthy that both biases are the same across the whole model, which incurs negligible storage overheads. Setting per-layer biases is left as future work.

## 5 EVALUATIONS

**Baselines and Network Architectures.** We compare BSFP with MSFP Darvish Rouhani et al. (2020), DSQ Nagel et al. (2019), TFlite Krishnamoorthi (2018), and APoT Li et al. (2020) in accuracy-to-precision trade-off. Furthermore, we also compare area and power efficiency with other strong baselines, such as BF16 Jouppi et al. (2020), fixed-point Nagel et al. (2019) and Power-of-Two PEs Zhou et al. (2017); Li et al. (2020).

We select six mainstream DNNs for evaluating BSFP and MSFP. ShuffleNet-v2 Ma et al. (2018) and MobileNet-v2 Sandler et al. (2018) are chosen to represent compact DNNs. ResNet-18 and ResNet-50 He et al. (2015) are selected to represent classical DNNs. EfficientNet-v2 Tan & Le (2019) and VIT Dosovitskiy et al. (2020) are chosen to represent modern DNNs. All of the results are based on ImageNet classification dataset Deng et al. (2009).

**Hardware Evaluations.** We implement the BSFP PE and baseline PEs in Verilog and validate the behavior against the software functional simulator. All designs are synthesized at 500 MHz under TSMC $40nm$ using Synopsys Design Compiler (Topographical mode). We set the parallelism to be 16 for all number systems.

### 5.1 MODEL SIZE-TO-ACCURACY AND VECTOR LENGTH-TO-MODEL SIZE TRADE-OFFS

Figure 5(a) shows the storage-to-accuracy Pareto frontier of BSFP and MSFP on ShuffleNet-v2 (post-training quantization). BSFP consistently improves the accuracy using a smaller model size. For example, BSFP simultaneously obtains 0.4% top-1 accuracy gain and saves 5% model size over MSFP. The superiority of BSFP is consistent across four evaluated models, where other results are omitted for brevity.

Figure 5(b) compares the accuracy-to-storage trade-offs of different vector lengths for BSFP and MSFP on ShuffleNet-v2 (post-training quantization). On BSFP, the vector length ranging from 16 to 64 leads to only a marginal accuracy drop, while a vector length of 128 significantly decreases the accuracy. Thus, we suggest that the vector length for BSFP be 16 to 64. In comparison, MSFP's accuracy drops significantly from a length of 16 to 32, and the accuracy does not vary significantly while further enlarging the vector length. It is also noteworthy that BSFP consistently outperforms the Pareto frontier regardless of vector length compared to MSFP.

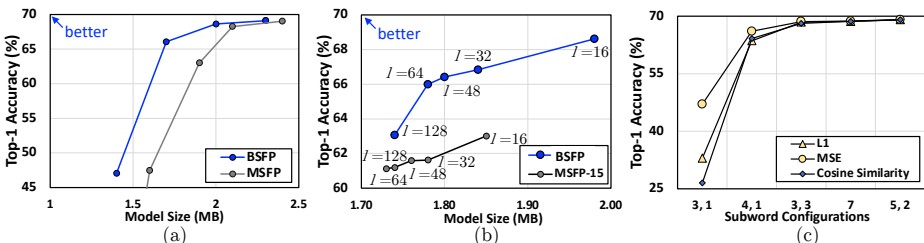

Figure 5: (a) Model size-to-accuracy Pareto frontiers of BSFP and MSFP with various bitwidths (fixing vector length). (b) Model size-to-accuracy Pareto frontiers of BSFP and MSFP with various vector length, $l$ (fixing bitwidth). (c) Quantization using different criteria. All experiments are conducted on ShuffleNet-v2 using post-training quantization.

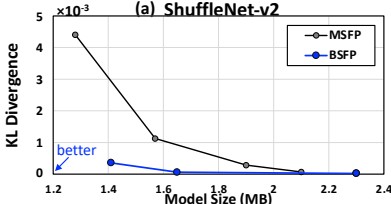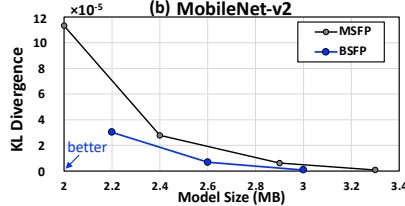

Figure 6: KL Divergence-to-model size comparison of MSFP and BSFP on (a) ShuffleNet-v2 and (b) MobileNet-v2. The KLD of BSFP is consistently lower than MSFP while using a smaller model size.

## 5.2 QUANTIZING USING DIFFERENT CRITERION

Figure 5(c) shows the accuracy of quantizing ShuffleNet-v2 using different criteria, i.e., L1 distance, MSE, and Cosine Similarity. The MSE is the best criterion in our grid search-based quantization flow. It is noteworthy that different criteria have little accuracy difference in higher bitwidth and vary considerably in lower bitwidths. One reason for MSE's superiority is that it punishes significant element-wise distortion, which the other two criteria cannot. Specifically, selecting L1 norm optimizes overall distortion, and selecting Cosine Similarity optimizes the angular difference.

## 5.3 KULLBACK-LEIBLER DIVERGENCE ANALYSIS

Kullback-Leibler Divergence (KL Divergence) is chosen for evaluating quantization quality in the MSFP paper. Intuitively, lower KL Divergence demonstrates the quantization can fit the original data much better, which results in higher accuracy. We compare the KL Divergence of MSFP and BSFP to demonstrate the superiority of BSFP.

Figure 6(a) and Figure 6(b) respectively sample a layer from ShuffleNet-v2 and MobileNet-v2 to compare the KL Divergence-to-model size Pareto frontier of BSFP and MSFP. The key takeaway is that BSFP consistently obtains better KL Divergence for both models while allocating smaller model sizes. Furthermore, the discrepancy between MSFP and BSFP enlarges as the model size reduces.

## 5.4 PER-VECTOR ABSOLUTE PEARSON'S SKEWNESS ANALYSIS

Figure 7 analyzes the per-vector absolute Pearson's skewness coefficient ($SK$) for each layer of ShuffleNet-v2 and ResNet-18. We report the proportion of vectors with different degrees of skewness, i.e., moderate ($0.5 < SK < 1.0$), high ($1.0 < SK < 1.5$), and very high ($1.5 < SK$). Two takeaways are: 1) The weight vectors on modern networks, i.e., ShuffleNet-v2 and ResNet-18, are skewed. Specifically, up to $50\% \sim 75\%$ of the weight vectors are skewed. The proposed BSFP can adapt to skewed distributions and outperform prior works with uniform quantization. 2) Even on weight vectors that are less skewed, BSFP can still outperform prior works because it does not waste quantization levels and provides better step size flexibility.

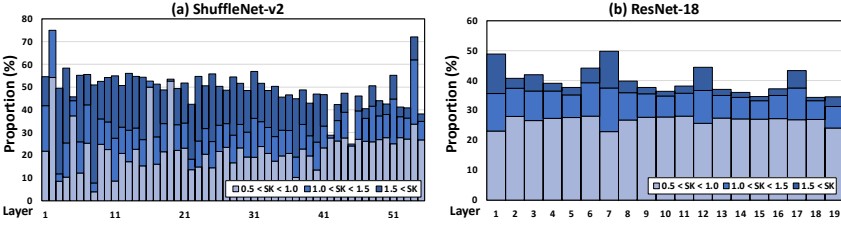

Figure 7: Per-vector absolute Pearson's skewness coefficient ($SK$) of (a) ShuffleNet-v2 and (b) ResNet-18. In general, very highly skewed means $SK > 1.5$, highly skewed means $SK > 1.0$, and moderately skewed means $1.0 > SK > 0.5$. Appendix G shows that slicing weight into vectors is one potential rationale for the large skewness.

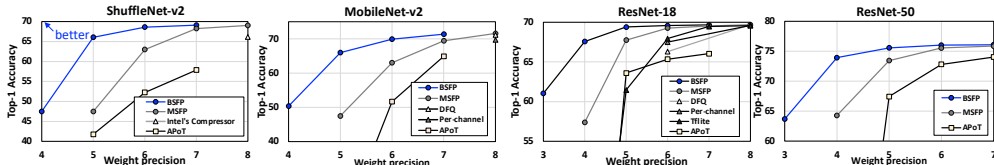

Figure 8: Precision-to-Accuracy comparison of BSFP with MSFP, DFQ, TFlite, APoT. All number systems utilize post-training quantization. The accuracy of DFQ Nagel et al. (2019) and TFlite Krishnamoorthi (2018) are derived from the papers. We implement APoT to obtain the accuracy.

Table 1: Post-training comparison of accuracy and hardware performance on ImageNet with MSFP and BSFP.

|  | Method | Precision (W / A) | Top-1 (%) | Model Size | FLOPs/ FixOPs (8b) | Method | Precision (W / A) | Top-1 (%) | Model Size | FLOPs/ FixOPs (8b) |
|---|---|---|---|---|---|---|---|---|---|---|
| **ShuffleNet-v2** | FP | 32 / 32 | 69.18 | 9.1 MB | 148.8M | | | | | |
| | MSFP | 8 / 8 | 69.03 | 2.4 MB | 148.8M | BSFP (OURS) | 7 [5+2] / 8 | 69.10 | 2.3 MB | 130.2M |
| | | 7 / 7 | 68.27 | 2.1 MB | 113.9M | | 6 [3+3] / 6 | 68.62 | 2.0 MB | 83.7M |
| | | 6 / 6 | 63.01 | 1.9 MB | 83.7M | | 5 [4+1] / 4 | 66.06 | 1.7 MB | 46.5M |
| | | 5 / 5 | 47.49 | 1.6 MB | 58.1M | | 4 [3+1] / 4 | 47.50 | 1.4 MB | 37.2M |
| **MobileNet-v2** | FP | 32 / 32 | 71.84 | 14.0 MB | 314.1M | | | | | |
| | MSFP | 8 / 8 | 71.58 | 3.7 MB | 314.1M | BSFP (OURS) | 7 [4+3] / 8 | 71.37 | 3.5 MB | 274.8M |
| | | 7 / 7 | 69.42 | 3.3 MB | 240.5M | | 6 [4+2] / 6 | 69.93 | 3.0 MB | 176.7M |
| | | 6 / 6 | 63.03 | 2.9 MB | 176.7M | | 5 [4+1] / 4 | 66.01 | 2.6 MB | 98.2M |
| | | 5 / 5 | 47.49 | 2.4 MB | 122.7M | | 4 [2+2] / 4 | 50.35 | 2.2 MB | 78.5M |
| **ResNet-18** | FP | 32/32 | 69.76 | 46.8 MB | 1.82G | | | | | |
| | MSFP | 8 / 8 | 69.69 | 12.4 MB | 1.82G | BSFP (OURS) | 7 [5+2] / 8 | 69.67 | 11.6 MB | 1.59G |
| | | 7 / 7 | 69.54 | 11.0 MB | 1.39G | | 6 [4+2] / 6 | 69.58 | 10.1 MB | 1.02G |
| | | 6 / 6 | 69.27 | 9.5 MB | 1.02G | | 5 [3+2] / 4 | 69.40 | 8.7 MB | 568.8M |
| | | 5 / 5 | 67.75 | 8.0 MB | 710.9M | | 4 [2+2] / 4 | 67.57 | 7.2 MB | 455.0M |
| | | 4 / 4 | 57.40 | 6.6 MB | 455.0M | | 3 [2+1] / 3 | 61.04 | 5.8 MB | 255.9M |
| **ResNet-50** | FP | 32/32 | 76.13 | 102.2 MB | 4.14G | | | | | |
| | MSFP | 8 / 8 | 76.06 | 27.2 MB | 4.14G | BSFP (OURS) | 7 [5+2] / 8 | 76.08 | 25.4 MB | 3.6G |
| | | 7 / 7 | 75.86 | 24.0 MB | 3.17G | | 6 [4+2] / 6 | 76.02 | 22.2 MB | 2.3G |
| | | 6 / 6 | 75.54 | 20.8 MB | 2.3G | | 5 [3+2] / 4 | 75.57 | 19.0 MB | 1.6G |
| | | 5 / 5 | 73.45 | 17.6 MB | 1.6G | | 4 [3+1] / 4 | 73.93 | 15.8 MB | 1.0G |
| | | 4 / 4 | 64.28 | 14.4 MB | 1.0G | | 3 [2+1] / 3 | 63.68 | 12.6 MB | 582.2M |
| **EfficientNet-v2 (s)** | FP | 32/32 | 84.23 | 88.0 MB | 8.8G | | | | | |
| | MSFP | 8 / 8 | 84.09 | 23.4 MB | 8.8G | BSFP (OURS) | 7 [5+2] / 8 | 84.06 | 21.8 MB | 7.7G |
| | | 7 / 7 | 83.94 | 20.6 MB | 6.7G | | 6 [4+2] / 6 | 83.96 | 19.1 MB | 5.0G |
| | | 6 / 6 | 83.08 | 17.9 MB | 5.0G | | 5 [4+1] / 6 | 83.12 | 16.3 MB | 4.2G |
| | | 5 / 5 | 77.45 | 15.1 MB | 3.4G | | 4 [2+2] / 4 | 76.08 | 13.6 MB | 2.2G |
| | | 4 / 4 | 3.27 | 12.4 MB | 2.2G | | 3 [2+1] / 4 | 27.20 | 10.8 MB | 1.7G |
| **Vision Transformer (ViT-B/16)** | FP | 32/32 | 81.07 | 348.1 MB | 56.0G | | | | | |
| | MSFP | 8 / 8 | 81.01 | 92.4 MB | 56.0G | BSFP (OURS) | 7 [5+2] / 8 | 80.92 | 86.3 MB | 49.0G |
| | | 7 / 7 | 80.94 | 81.6 MB | 42.9G | | 6 [4+2] / 6 | 80.87 | 75.4 MB | 31.5G |
| | | 6 / 6 | 80.84 | 70.7 MB | 31.5G | | 5 [4+1] / 4 | 80.41 | 64.6 MB | 17.5G |
| | | 5 / 5 | 80.13 | 59.8 MB | 21.9G | | 4 [2+2] / 4 | 80.21 | 53.7 MB | 14.0G |
| | | 4 / 4 | 76.89 | 48.9 MB | 14.0G | | 3 [2+1] / 4 | 79.77 | 42.8 MB | 10.5G |

## 5.5 WEIGHT PRECISION-TO-ACCURACY PARETO FRONTIER

Figure 8 compares the weight precision-to-accuracy Pareto frontier of MSFP and BSFP on four DNNs. BSFP consistently outperforms MSFP on accuracy, given the same weight precision. Furthermore, the accuracy gap between MSFP and BSFP enlarges while reducing the weight precision. Let us take ShuffleNet-v2 as an example. When allocating 7b weight precision for both BSFP and MSFP, BSFP achieves 69.10% top-1 accuracy, outperforming MSFP's 68.27% accuracy by 0.83%. However, when further narrows down the weight bit to five, the accuracy benefit of BSFP over MSFP enlarges. Further, BSFP consistently outperforms MSFP and other numerical systems under the same precision.

Table 1 summarizes the accuracy, model size, and the number of operations on six DNNs. For full precision models, we report their computation complexity in FLOPs. We define a FixOP as one operation between an 8-bit fixed-point weight and an 8-bit fixed-point activation, which takes 64 binary operations for quantized models. For example, a 4 by 6-bit multiplication is equivalent to $^{4\times6}/_{8\times8} = ^3/_8$ FixOP (8b). To sum up, BSFP consistently achieves higher accuracy with smaller model size and computation complexity than MSFP. Smaller FixOPs improve the computation steps on serial PE or reduce hardware overheads on parallel PE.

Table 2: BSFP versus other mainstream number formats for DNN inference. Memory and MAC density of various formats are normalized to BF16. The results listed are based on topographical synthesis results using TSMC $40nm$ process at 500 MHz.

| | 16× BF16 | 16× INT4 | 16× INT8 | 16× Power-of-Two | Bit-Serial MSFP PE | Scaled Serial PE (BSFP) | |
| --- | --- | --- | --- | --- | --- | --- | --- |
| | | | | | | 2b-mult | 1b-mult |
| Area (per PE) | 1.0× | 22.6× | 8.7× | 7.8× | 13.0× | 26.6× | 33.1× |
| Power (per PE) | 1.0× | 13.0× | 5.1× | 4.8× | 8.5× | 14.1× | 24.6× |

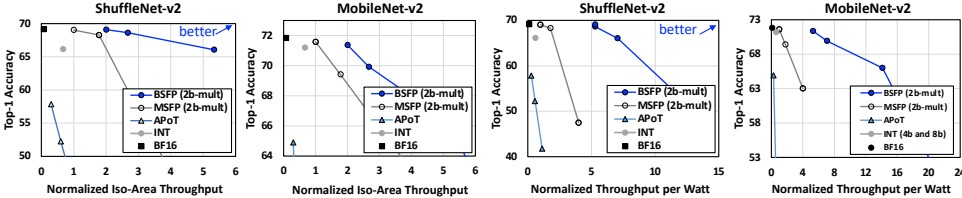

Figure 9: Iso-area Throughput to Accuracy (left) and Throughput per Watt to Accuracy (right) comparison of BSFP with other strong baselines. We normalize the throughput and throughput/W to MSFP-16.

## 5.6 Accuracy to Performance and Accuracy to Energy Efficiency Comparisons

Table 2 shows the area and power of PEs with different number formats. We normalize all of the designs with BF16 PE. In general, scaled serial PE achieves the smallest area and power because of the compact MAC circuit. The 2-b scaled serial PE can outperform 2-b serial MSFP PE Qian Zhang et al. (2022) because of the following reasons:

- Larger adder tree induced by additional XOR gate: Besides area overheads of XOR gates, the bitwidth of the multiplication results must be enlarged because of the potential negation. Consequently, the 16×4-b adder tree has to increase each input port by 1-b, causing 25% higher complexity than the original adder tree. In brief, the XOR gates, the negation logic, and the enlarged adder tree jointly cause the area and energy in-efficiency.

- The alignment overheads for MSFP are more significant than for BSFP: Although the bitwidth of scaling factors are similar for BSFP and MSFP, MSFP allocates all 8-b to be the exponent. In contrast, BSFP allocates only 3-b to be exponent and lets the rest be sign and mantissa. As a result, the alignment procedure before accumulation is largely simplified for BSFP, which improves energy efficiency.

Figure 9 (left) shows the iso-area throughput to accuracy comparison. BSFP consistently outperforms MSFP across four models. Specifically, BSFP outperforms MSFP by 1.3× to 2.0× throughput while achieving higher accuracy. Again, the throughput improvement results from a smaller PE area and lower weight precision. Figure 9 (right) shows the throughput per W to accuracy comparison. BSFP also achieves significantly better throughput per Watt than MSFP. Specifically, BSFP outperforms MSFP by 1.3× to 5.3× while achieving higher accuracy.

## 6 Conclusions

This paper introduces Block and Subword-Scaling Floating-Point (BSFP) for quantizing weight vectors in neural networks, which typically exhibit a skewed and non-uniform distribution. The quantized vector of BSFP is the sum of a set of subword-scaling vectors, bringing up 2× speedup and 5.3× energy efficiency improvements compared with MSFP.

In addition, our grid search method guarantees finding the optimal BSFP configurations for capturing skewed and non-uniform weights and optimizing the MSE criterion. With our proposed scaled serial PE, BSFP obtains better area efficiency and energy efficiency compared to prior number systems. In sum, BSFP successfully reaches state-of-the-art model size, throughput, and energy efficiency.

ACKNOWLEDGMENTS

We thank reviewers for their insightful comments. We thank NCHC (National Center for High-performance Computing, Taiwan) and TSRI (Taiwan Semiconductor Research Institute) for providing computational and storage resources. This work is supported in part by NSTC (National Science and Technology Council, Taiwan) grants 111-2823-8-007-001 and 111-2218-E-007-009, Synopsys Scholarship, and NSTC Scholarship.

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

# APPENDICES

## A    QUANTIZATION-AWARE FINE-TUNING (QAT)

**Fine-tuning Setups.**    BSFP can also be applied to quantization-aware fine-tuning. The fine-tuning procedure chooses SGD as the optimizer and sets the learning rate to be $10^{-6}$. In addition, we set the momentum to 0.9 and weight decay to $10^{-5}$. The models are fine-tuned on NVIDIA's V100 GPUs using a batch size of 100. We extend Pytorch to conduct the algorithmic experiments.

Table 3 shows the quantize-aware fine-tuning comparison between MSFP and BSFP for wide range of models. The key takeaway is:

- **Consistency of the benefits:** BSFP consistently achieves higher accuracy using smaller model size and fewer operations. This demonstrates that the benefits of BSFP is consistent regardless of post-training quantization or quantize-aware training. Further, we also point out that BSFP outperforms MSFP in accuracy given ultra low bitwidth.

Table 3:  Quantization-aware fine-tuning comparison of accuracy performance as well as hardware performance of ShuffleNet-v2 Ma et al. (2018), MobileNet-v2 Sandler et al. (2018), ResNets He et al. (2015) on ImageNet with MSFP and BSFP.

| | Method | Precision (W / A) | Top-1 (%) | Model Size | FLOPs/ FixOPs (8b) | Method | Precision (W / A) | Top-1 (%) | Model Size | FLOPs/ FixOPs (8b) |
|---|---|---|---|---|---|---|---|---|---|---|
| ShuffleNet-v2 | FP | 32 / 32 | 69.18 | 9.1 MB | 148.8M | | | | | |
| | | 8 / 8 | 68.89 | 2.4 MB | 148.8M | | | | | |
| | MSFP | 7 / 7 | 68.30 | 2.1 MB | 113.9M | BSFP (OURS) | 7 [5+2] / 8 | **69.10** | 2.3 MB | 130.2M |
| | | 6 / 6 | 67.56 | 1.9 MB | 83.7M | | 6 [3+3] / 6 | **68.63** | 2.0 MB | 83.7M |
| | | 5 / 5 | 64.17 | 1.6 MB | 58.1M | | 5 [4+1] / 4 | **65.69** | 1.7 MB | 46.5M |
| | | 4 / 4 | 44.99 | 1.3 MB | 37.2M | | 4 [3+1] / 4 | **62.73** | 1.4 MB | 37.2M |
| MobileNet-v2 | FP | 32 / 32 | 71.84 | 14.0 MB | 314.1M | | | | | |
| | | 8 / 8 | 71.71 | 3.7 MB | 314.1M | | | | | |
| | MSFP | 7 / 7 | 71.19 | 3.3 MB | 240.5M | BSFP (OURS) | 7 [5+2] / 8 | **71.80** | 3.5 MB | 274.8M |
| | | 6 / 6 | 70.27 | 2.9 MB | 176.7M | | 6 [3+3] / 6 | **71.17** | 3.0 MB | 176.7M |
| | | 5 / 5 | 65.25 | 2.4 MB | 122.7M | | 5 [4+1] / 4 | **67.38** | 2.6 MB | 98.2M |
| | | 4 / 4 | 45.73 | 2.0 MB | 78.5M | | 4 [2+2] / 4 | **66.29** | 2.2 MB | 78.5M |
| ResNet-18 | FP | 32/32 | 69.76 | 46.8 MB | 1.82G | | | | | |
| | | 8 / 8 | 69.84 | 12.4 MB | 1.82G | | | | | |
| | MSFP | 7 / 7 | 69.73 | 11.0 MB | 1.39G | BSFP (OURS) | 7 [5+2] / 8 | **70.01** | 11.6 MB | 1.59G |
| | | 6 / 6 | 69.64 | 9.5 MB | 1.02G | | 6 [3+3] / 6 | **69.85** | 10.1 MB | 1.02G |
| | | 5 / 5 | 68.94 | 8.0 MB | 710.9M | | 5 [3+2] / 4 | **69.09** | 8.7 MB | 568.8M |
| | | 4 / 4 | 64.76 | 6.6 MB | 455.0M | | 4 [2+2] / 4 | **69.02** | 7.2 MB | 455.0M |
| | | 3/ 3 | 46.59 | 5.1 MB | 255.9M | | 3 [2+1] / 3 | **64.85** | 5.8 MB | 255.9M |
| ResNet-50 | FP | 32/32 | 76.13 | 102.2 MB | 4.14G | | | | | |
| | | 8 / 8 | 76.26 | 27.2 MB | 4.14G | | | | | |
| | MSFP | 7 / 7 | 76.17 | 24.0 MB | 3.17G | BSFP (OURS) | 7 [5+2] / 8 | **76.30** | 25.4 MB | 3.6G |
| | | 6 / 6 | 76.06 | 20.8 MB | 2.3G | | 6 [4+2] / 6 | **76.17** | 22.2 MB | 2.3G |
| | | 5 / 5 | 75.12 | 17.6 MB | 1.6G | | 5 [4+1] / 4 | **76.00** | 19.0 MB | 1.6G |
| | | 4 / 4 | 71.57 | 14.4 MB | 1.0G | | 4 [2+2] / 4 | **75.04** | 15.8 MB | 1.0G |
| | | 3 / 3 | 53.47 | 11.2 MB | 582.2M | | 3 [2+1] / 3 | **70.95** | 12.6 MB | 582.2M |

## B    COMPARISON BETWEEN STANDARD QAT AND THE PROPOSED LOW-COST QAT

To amortize the overheads incurred by grid search during quantization-aware training (QAT), we propose to quantize only the first batch for every 100 batches to amortize the quantization overheads. We refer this proposal as low-cost QAT.

To answer the question whether the proposed low-cost QAT will affect the training process, Figure 10 compares the training curve of the standard QAT and low-cost QAT on fine-tuning ShuffleNet-v2 and MobileNet-v2. The takeaway is that the training curve of low-cost QAT is similar to the curve

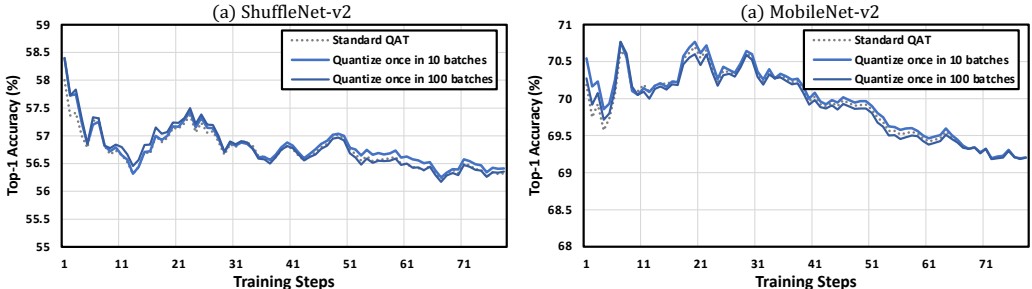

Figure 10: Training curve comparison between standard quantization-aware training and low-cost quantization-aware training for (a) ShuffleNet-v2 and (b) MobileNet-v2.

standard QAT, which quantizes before every batch. This demonstrates that the low-cost QAT can preserve the training curve while requiring significantly lower search overheads.

## C  EVALUATIONS ON OTHER TASKS

### C.1  OPTICAL FLOW ESTIMATION USING RAFT ON KITTI DATASET

Since our research direction focuses on computer vision, we choose to extend evaluate for BSFP on DNN-based optical flow estimation Teed & Deng (2021) (KITTI dataset Menze & Geiger (2015)). Table 4 shows that BSFP consistently outperforms MSFP using same weight precision, which demonstrates the benefit of BSFP is consistent regardless of the tasks.

Table 4: MSFP and BSFP on RAFT Teed & Deng (2021) for KITTI dataset Menze & Geiger (2015). BSFP consistently improves the error given same weight precision.

| Optical Flow (RAFT) | MSFP | BSFP |
|---|---|---|
| Precision (W) | Average end-point error | |
| 9 | 0.658 | 0.632 |
| 7 | 0.895 | 0.682 |
| 5 | 2.068 | 0.905 |

## D  DETAILS OF THE CRITERION-OPTIMAL QUANTIZATION FLOW

In order to find the optimal subword mantissas and scalings, we adopt iterative rounding, which contains the following steps:

1. Enumerate the two scaling factors.
2. List the quantization levels the two scaling factors can generate. The number of quantization levels is moderate. For example, there are only 64 levels if the two subwords are 2+4 bits.
3. Calculate the MSE between the 16 original weights and their nearest quantization levels. Finding the nearest quantization level in a list is viable as other quantization schemes, such as APoT Li et al. (2020), also employ it.
4. Keep the scaling factors that achieve the lowest MSE and go to step 1.

The pseudo code of the criterion-optimal quantization flow is presented in Algorithm 1

## E  SUPPORTING BSFP ON BIT-PARALLEL ARCHITECTURE

Since some of the hardware, e.g., CPU and GPU, prefers bit-parallel PE architecture, Figure 11 presents the bit-parallel BSFP PE design. The difference between serial PE architecture and parallel

---

**Algorithm 1** Criterion-optimal quantization flow

---

**Input:** Full-precision weight $W_{fp}$
**Output:** Criterion-optimal scalings $sc_{opt}$, Criterion-optimal subword mantissas $man_{opt}$
**Require:** Subword configurations $config_{sub}$, Scaling configurations $config_{sc}$

$\quad Criterion \leftarrow \max$
$\quad SC \leftarrow Enum(config_{sc})$        ▷ Enumerate all scaling setups to form search space $SC$
$\quad$**for** $sc_{cur} \in SC$ **do**        ▷ For every $sc_{cur}$ of the search space $SC$
$\quad\quad list_{quant} \leftarrow BuildQlist(sc_{cur}, config_{sub})$    ▷ Find all quantization levels using $sc_{cur}$
$\quad\quad W_{quant} \leftarrow Quant(W_{fp}, list_{quant})$    ▷ Quantize $W_{fp}$ to its nearest levels in list
$\quad\quad Criterion_{cur} \leftarrow MSE(W_{fp}, W_{quant})$    ▷ Compute the MSE criterion
$\quad\quad$**if** $Criterion_{cur} < Criterion$ **then**.    ▷ Update the scale and subword mantissas
$\quad\quad\quad sc_{opt} \leftarrow sc_{cur}$
$\quad\quad\quad man_{opt} \leftarrow man_{cur}$
$\quad\quad\quad Criterion \leftarrow Criterion_{cur}$
$\quad\quad$**end if**
$\quad$**end for**

---

PE architecture is that the scaling units need to be unrolled, which incurs some overheads. BSFP's benefits of smaller model size and lower arithmetic complexity remain unchanged.

Further, we can achieve uneven subword configurations by fixing the upper multipliers and adjusting only the weight precision of lower multipliers (red weights).

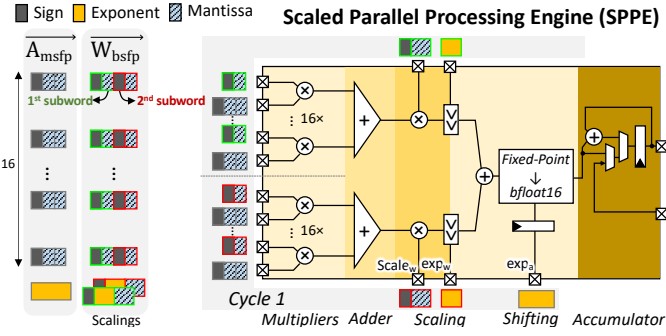

Figure 11: Proposed scaled parallel processing engine.

BSFP can be efficiently deployed in newly-manufactured devices, including customized AI accelerator IPs, which represent a significant amount of AI products including smartphones and smart appliances. For example, Nvidia NVDLA NVIDIA (2023), Qualcomm Hexagon Tensor Accelerator Codrescu (2013), and MIT Eyeriss Chen et al. (2017). Such IPs are embedded in emerging AI chips for smartphones, appliances, robots, etc. Since customized, embedded AI accelerator IPs are not obligated to support standard data types, the proposed BSFP type, which improves weight quantization quality (including post-training quantization, which is even difficult and desirable), will stand out as a candidate for chip designers to consider and adopt in their new design.

As for CPUs, although they are not our only focus, letting newly-manufactured CPUs support BSFP is not difficult, and the computation will be efficient. Modern CPUs, including RISC-V, ARM, and Synopsys ARC, allow custom ISA extensions Imperas (2019); Arm (2020); Synopsys (2020). Designers can provide an add-on digital circuit that natively and efficiently supports BSFP using low-bit, fixed-point adders and multipliers. Vendors already offered the required design/simulation flow, CPU generator, and toolchain generator to facilitate designers to employ the extended instructions, e.g., vector product instructions between BSFP weights and MSFP activations.

## F  DETAILS OF THE HARDWARE COMPARISON

**Processing Engine Details:** Table 2 compares 16-wide PEs in different number formats, i.e., **BF16, INT4, INT8, Power-of-Two, MSFP, and BSFP**, following similar setup in FAST Qian Zhang et al. (2022). The architectural details for these PEs are summarized below:

- The **BF16 (brain float)** PE adopts bit-parallel architecture, which computes 16-wide BF16 multiplications and reduces them to a single partial sum. The accumulator is in BF16 format.
- The **INT4** and **INT8** PEs also adopt bit-parallel architectures, which perform 16-wide MAC and generate 12b and 20b partial sum, respectively. The accumulator then accumulates it using INT32 format.
- The **Power-of-Two** PE shifts 16 inputs based on the weight values and reduces 16 products to a single partial sum. The accumulator is in INT32 format.
- The serial **MSFP** PE Qian Zhang et al. (2022) computes 16 2b-to-2b multiplications per cycle and shifts them to accumulate for the correct partial sum. We adopt the BF16 accumulator.
- The proposed $S^2IP$ supports **BSFP**, which computes 16-wide 2b or 1b multiplications per cycle and scale-and-shifts them to accumulate for correct partial sum. The accumulator is in BF16 format.

Additional Notes:

- The adder tree output precisions of every PEs are optimized to their minimum.
- ALL of the PEs obtain similar slack profiles.

**Performance Analysis Setup:** We adopt iso-area performance setup to fairly compare different number systems, which is widely adopted by top architecture papers Sharify et al. (2019); Yang et al. (2021). In other words, if the size of BF16 PE is 26× larger than that of the BSFP PE, we are allowed to allocate 26× more PEs for BSFP. Besides PE parallelism, we also consider the multi-cycle computation of serial architecture to reasonably estimate the performance.

**Power Analysis Setup:** We implement all of the above PEs using Verilog and synthesis them using Synopsys Design Compiler (topographical mode) on the TSMC 40 nm node. Specifically, the power of each module is analyzed with average switching activity. We then utilize the power profiles to estimate throughput per Watt.

## G  PER-LAYER SKEWNESS ANALYSIS

Figure 12 analyzes the absolute skewness for weights of each layer without slicing weights into vectors. We observe that most layers' skewness is low (i.e., ≤0.5), which matches the summary of prior works that weight values follow bell-shape and non-skewed distribution.

Through a side-by-side comparison between Figure 12 and Figure 13 (same as Figure 7; We copy it here for easier comparison), we clearly observe that vectorization is one source of more considerable skewness on Figure 13.

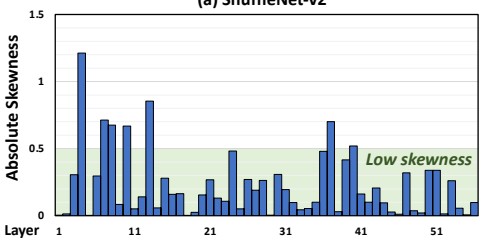 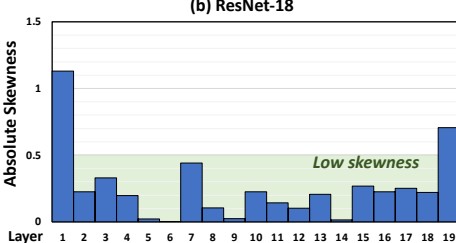

Figure 12:  Per-layer absolute skewness of (a) ShuffleNet-v2 and (b) ResNet-18. Most of the layers obtain low skewness, which matches the analysis reported by prior works.

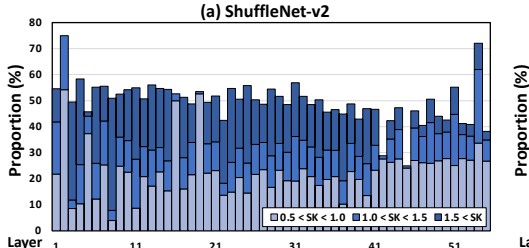 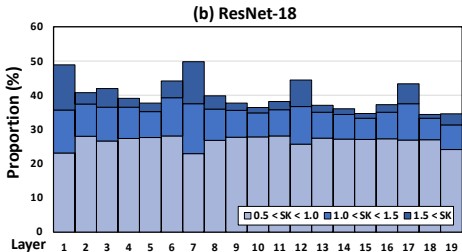

Figure 13: Per-vector absolute Pearson's skewness coefficient ($SK$) of (a) ShuffleNet-v2 and (b) ResNet-18. In general, very highly skewed means $SK > 1.5$, highly skewed means $SK > 1.0$, and moderately skewed means $1.0 > SK > 0.5$.

## H    SEARCHING FOR BSFP CONFIGURATIONS

To obtain the scaling factor configurations (1-4-3, 1-3-3) and the exponent biases (-8, -3), we use the following steps to empirically set them. This method is acceptable as neural networks naturally posses many hyper parameters.

1. Begin with setting scaling factors to BF16 (1-7-8), a sufficiently precise format.

2. Given the scaling factors, find the narrowest bitwidths of subwords that causes an accuracy drop less than a first budget. For example, 5+2 bits are selected.

3. Given the subword settings, reduce the bitwidths of scaling factors and sweep the the biases to a point that the overall accuracy drop is less than a second budget. This step lead us to (1-4-3, 1-3-3) and biases (-3,-8).

## I    ADDITIONAL COMPARISON OF BSFP AND MSFP (BFP)

Compared to Vanilla BFP (MSFP), BSFP lowers the flexibility converting from full-precision weights, but BSFP gains richer flexibility to approximate and convert back to full-precision weights. In view of the different characteristics of weights and activations, it should be a reasonable option to adopt BSFP for weights and MSFP for activations.

BSFP focuses on fine-tuning-free, post-training quantization (as well as inference), which are our heard realistic demands from the industry. Though BSFP-based training is beyond the scope, here we provide a proposal.

First, each BSFP vector is a weighted sum of two Vanilla BFP vectors. Therefore, by letting the scaling factors be trainable parameters, it is possible to leverage a Vanilla BFP framework to train BSFP networks in the first place without resorting to decomposition-based quantization.

Second, one can employ Vanilla BFP training for the majority of training epochs in the beginning and switch to BSFP training only at the last few epochs.

