# OpenReview forum: "Block and Subword-Scaling Floating-Point (BSFP) : An Efficient Non-Uniform Quantization For Low Precision Inference"
_ICLR.cc/2023/Conference — ICLR 2023 poster_

### Official Review · Reviewer_h3B6 · 2022-10-23

**Confidence:** 4
**Correctness:** 3
**Technical Novelty And Significance:** 3
**Empirical Novelty And Significance:** 3
**Recommendation:** 6

**Clarity, Quality, Novelty And Reproducibility:**

The paper is well written and for the most part easy to follow.
As far as I am aware, the technique is novel insofar the split of input vectors into subwords is concerned.
It describes the proposed format and process flow in detail, allowing for reproducibility at SW level. The HW simulation part severely lacks information though.

**Strength And Weaknesses:**

Strengths:
- the idea is conceptually simple and well-described: implementing non-uniform weight coverage by combining two (or potentially more) scales for weight quantization
- comparison against MSFP competing format is favorable, showing improved accuracy at given model size across a range of networks for image classification
- practical implementation is supported by a proposed HW design and process flow for the processing engine / architecture

Weaknesses and other comments:
- there are several empirical choices across the paper: scaling factor format (1-4-3, 1-3-3), biases (-3,-8), MSE. For the most part, it is not clear what considerations drove these selection, what other options were investigated, and what was their impact. As far as MSE is concerned, alternative criterions are shown in Fig. 5(c) but the authors don't even attempt to explain why MSE may perform better than L1 or cosine similarity
- this technique explores all possible scaling factors combinatorially in search of the optimal one. This seems feasible for a small network but less practical for larger ones. Can the authors mention the search time required for the other networks they investigated? In addition, the number of possibilities is upper bounded to $2^7 * 2^7$. Is this value compounded by the number of subword configurations to explore (therefore larger for larger bit widths)?
- no details on HW simulation makes any sort of verification of throughput and energy efficiency impossible. These may warrant their own section in the appendix
- accuracy improvement at given model size or FLOPs can be modest in some cases (Table 1), although it is general more apparent when quantization is more aggressive
- section 5.4 presents skewness analysis to support the claim that distributions of grouped weights can be skewed. I assume Fig.7 was done for vector length of 16. Is the skewness arising from the grouping of weights into blocks or from the actual skewness of the weight distribution of each layer? What would be the proportion of skewed vectors taken from a normally distribution? May want to present this as "baseline" skewness

Other comments:
- clarification on Table 1: when reporting weight bit precision for BSFP (for example: 7 [5 + 2]) does this mean the input vectors were split into 2 subwords in 2s complements format and respective size 5 and 2? Are the scaling factors fixed to 1-4-3  and 1-3-3 format?
- regarding QAT in the appendix, how does the process work? Are the optimal scaling factors recomputed at every iterations (or every 10, 100 iterations in the case of low cost QAT)? This seems to be adding an enormous overhead to the training



**Summary Of The Paper:**

This paper introduces a new floating point format for non-uniform quantization of weights (BSFP). The main idea is to split the input vectors (words) into subwords (usually 2) to which separate scaling factors are assigned. All possible combinations of scaling factors are evaluated and one is chosen according to a given criterion (e.g., MSE). The focus of the paper is on post training quantization (PTQ) but quantization aware training (QAT) results are also included. Results are reported on various network for image classification. A hardware implementation for BSFP is proposed and simulated.

**Summary Of The Review:**

The paper presents a novel format for weight quantization and related HW implementation. The proposed format generally outperforms direct competitors in terms of accuracy at a given model size and/or FLOPs, as tested over a range of models for image classification. Some empirical choices may benefit for clearer motivation and/or explanation.

---

> ### Author Response · Authors · 2022-11-19
> **Response to Reviewer h3B6 [Part-1]**
>
> We appreciate your insightful questions and comments. Please let us address the questions and comments below.
>
> **[Weakness 1] There are several empirical choices across the paper: scaling factor format (1-4-3, 1-3-3), biases (-3,-8), MSE. For the most part, it is not clear what considerations drove these selection, what other options were investigated, and what was their impact.**
>
> Yes, we use the following steps to empirically set them. This method is acceptable as neural networks naturally possess many hyper parameters.
>
>  1. Begin with setting scaling factors to BF16 (1-7-8), a sufficiently precise format.
>  2. Given the scaling factors, find the narrowest bitwidths of subwords that cause an accuracy drop less than a first budget.  For example, 5+2 bits are selected.
>  3. Given the subword settings, reduce the bitwidths of scaling factors and sweep the biases to a point that the overall accuracy drop is less than a second budget. This step leads us to (1-4-3, 1-3-3) and biases (-3,-8).
>
> **[Weakness 2] As far as MSE is concerned, alternative criterions are shown in Fig. 5(c) but the authors don't even attempt to explain why MSE may perform better than L1 or cosine similarity.**
>
> One possible reason why MSE performs better than L1 and cosine similarity is because MSE tends to minimize absolute quantization errors and at the same time level the quantization errors among 16 weights.  In comparison, L1 tends to allow an outlier that exhibits significant quantization errors to exist. In addition, cosine similarity reflects vector proportionality but ignores absolute quantization errors. We have incorporated this explanation in Section 5.2.
>
> **[Question 1] The search-based technique explores all possible scaling factors combinatorially in search of the optimal one. This seems feasible for a small network but less practical for larger ones. Can the authors mention the search time required for the other networks they investigated?**
>
> First of all, we have additionally reported the search time of the two largest networks, Vision Transformer and ResNet-50 (page 5). Their search time is within 1.5 and 1 hour, respectively, on a single NVIDIA V100.
>
> The search is a one-time effort, so 1 or 1.5 hour is acceptable. In addition, weight quantization is easily parallelizable by dividing the weights to be quantized. Therefore, if necessary, one can employ multiple GPUs to speed up processing large networks.
>
> **[Question 2] The number of possibilities is upper bounded to 2^8*2^7. Is this value compounded by the number of subword configurations to explore (therefore larger for larger bit widths)?**
>
> We show the algorithm as follows and Appendix D.  Steps 2 and 3 are compounded by the number of subword bit widths, but Step 4, MSE calculation, is not.
> 1. Enumerate the two scaling factors.
> 2. List the quantization levels the two scaling factors can generate. The number of quantization levels is moderate. For example, there are only 64 levels if the two subwords are 2+4 bits.
> 3. Finding the nearest quantization levels for 16 original weights. Such table-based quantization is viable as other quantization schemes, such as APoT (ICLR 20), also employ it.
> 4. Calculate the MSE between the 16 original weights and the nearest quantization levels.
> 5. Keep the scaling factors that achieve the lowest MSE and go to step 1.
>
> **[Question 3] No details on HW simulation makes any sort of verification of throughput and energy efficiency impossible. These may warrant their own section in the appendix.**
>
> Thank you for the suggestion. We have added Appendix F to describe our hardware evaluation methodologies and setups, which are comparable to the related architecture papers, such as Laconic (ISCA 19) and FuseKNA (HPCA 21). Specifically, we realize MSFP, Power-of-Two, BF16, INT4, and INT8-based PEs in Verilog, synthesize the PEs using Synopsys Design Compiler with the TSMC 40 nm standard cell libraries at 500 MHz, and report per-PE area, per-PE power, and iso-area throughput according to the synthesis-stage results.  For example, Figure 9 (left) compares accuracy and iso-area throughput, which is a fair comparison.  Low-precision hardware hurts the accuracy but favors the iso-area throughput. BSFP's tradeoff curve is close to the upper-right corner the most and outperforms the other formats, such as MSFP.

---

> > ### Author Response · Authors · 2022-11-19
> > **Response to Reviewer h3B6 [Part-2]**
> >
> > **[Question 4] accuracy improvement at given model size or FLOPs can be modest in some cases (Table 1), although it is general more apparent when quantization is more aggressive**
> >
> > BSFP' advantages in accuracy and complexity are apparent in many cases. Let us take ShuffleNet and MobileNet (Figure 8 or Table 1), for example.  First, given 6-bit weights and 6-bit activations, BSFP achieves 5.62% and 6.9% higher accuracy than MSFP.  In other words, BSFP's accuracy gains are significant at 6-bit quantization, which is not aggressive.  This is also the case for 6-bit ResNet-50, ResNet-18, and EfficientNet. In addition, MSFP can close the accuracy gaps of ShuffleNet and MobileNet to 0.35% and 0.51% by employing 7-bit weights and 7-bit activations.  However, compared with BSFP's 6 by 6-bit multiplications, MSFP's multiplications is (7*7)/(6*6) = 36% more complex.
> >
> > **[Suggestion] Section 5.4 presents skewness analysis to support the claim that distributions of grouped weights can be skewed. I assume Fig.7 was done for vector length of 16. Is the skewness arising from the grouping of weights into blocks or from the actual skewness of the weight distribution of each layer? What would be the proportion of skewed vectors taken from a normally distribution? May want to present this as "baseline" skewness**
> >
> > Thank you for the suggestions. First, yes, Figure 7 was done for vector length of 16.  Second, we have included the "baseline (per-layer)" skewness in Appendix G and Figure 12, which shows that the skewness is arising from the grouping of weights into blocks.
> >
> > **[Comment 1] Clarification on Table 1: when reporting weight bit precision for BSFP (for example: 7 [5 + 2]) does this mean the input vectors were split into 2 subwords in 2s complements format and respective size 5 and 2? Are the scaling factors fixed to 1-4-3 and 1-3-3 format?**
> >
> > Yes, 7 [5 + 2] means input mantissa is split into two subwords in 2s complement, and respective sizes are 5 and 2 bits, and yes, the scaling factors are fixed to 1-4-3 and 1-3-3 format.
> >
> > **[Comment 2] Regarding QAT in the appendix, how does the process work? Are the optimal scaling factors recomputed at every iterations (or every 10, 100 iterations in the case of low cost QAT)? This seems to be adding an enormous overhead to the training.**
> >
> > In our experiments, we let the scaling factors be recomputed for every 100 batches (20,000 images) during the fine-tuning process to lower the overhead. Since this work focuses on fine-tuning-free, post-training quantization (as well as inference), which are our heard realistic demands from the industry, more studies on fine-tuning are beyond the scope and left as future works.

---

> > > ### Comment · Reviewer_h3B6 · 2022-12-02
> > > **update**
> > >
> > > I thank the authors for addressing my questions. Despite justified concerns related to adoption of the proposed technique, which requires specialized HW, I believe there is sufficient novelty in the proposed format and the promising results shown on a variety of networks warrant publication. I maintain my score and lean towards acceptance.

---

### Official Review · Reviewer_KfxR · 2022-10-27

**Confidence:** 4
**Correctness:** 3
**Technical Novelty And Significance:** 3
**Empirical Novelty And Significance:** 3
**Recommendation:** 6

**Clarity, Quality, Novelty And Reproducibility:**

The most direct antecedent to this work appears to be prior additive quantization works (e.g., Additive Quantization for Extreme Vector Compression, Babenko and Lempitsky, CVPR2014 comes to mind). The operating space here is quite different, being around both data compression and hardware efficiency of performing computation on the data, whereas the Babenko work was around compression of high dimensional vectors in CV, similarity search and the like. The additive quantization here differs being not whole vector quantization using a codebook of VQ centroids but a combination of per-scalar values with shared scaling factors. So on the quantization front I think this is not super novel but the existence proof of showing that this works in the extreme compression regime is very useful, not to mention that the design space of "additive quantization" is itself fairly huge.

Also, I think adapting it in this extreme compression case for HW NN inference is new and interesting, and offers another design axis to the existing NN compression literature. Just as product quantization offers efficiencies in computation with respect to inner product or L2 distance between compressed vectors (no need to perform the full arithmetic, just use a bunch of lookup tables), here the BSFP additive compression work drastically reduces the size/number of full-on FP adders/multipliers required.

The one thing that I would be interested in seeing would be a wider variety of experiments on non-vision networks (language models, recommendation MLPs, etc) as the application in industry for this technique I think would be limited unless there was greater confidence that the design was not overfitting on specifics of this small subset of NNs considered.

I might not understand all of the details presented (and if one were to implement it there might be some fine details missing from the paper) but I do feel like I understand the scheme and could likely implement something like it myself both in software simulation and in RTL using the idea as a starting point.


**Strength And Weaknesses:**

The main strength is in showing the empirical applicability of the BSFP technique to a variety of computer vision NNs, and while I am not super up on the universe of inference quantization techniques that have been proposed in recent years, I feel like I am fairly up on the computer arithmetic in silicon design space and this quantization method does provide significant advantages in accelerating computation.

The in-silicon hardware advantages of this design seem quite obvious to me and I don't doubt the PPA advantages at all, but a little more detail on the comparison baselines would be useful. In "Hardware Evaluations", it seems that synthesis was run on a 16-wide S2PE (guessing this is what is mean by "parallelism to be 16 for all number systems), but what exactly was the synthesized design for a 16-wide bf16, int4, int8 design? Does the 16-wide int8 inner product tree produce a 32-bit output or a 8 * 2 + log2(16) = 20 bit output? Where did the bf16 blocks come from (DesignWare or elsewhere)? Any information on the slack in the various designs were produced? How was power estimated, using average switching activity or by using realistic test vector SAIFs from real networks?

Table 1 is a little confusing to look at. The bolded numbers are for all top-1 results of BSFP, though some of the bolded numbers are under the neighboring MSFP numbers. However I would agree that the two techniques are not directly comparable (as BSFP does offer big HW efficiencies). The Pareto frontier charts are simply easier to understand. Regarding "FLOPs / FixOPs (8b)", how is this computed for BSFP as many of the adders / multipliers in the S2PE are sub-8b?

**Summary Of The Paper:**

The authors propose a form of low-precision additive quantization in order to represent weights/parameters in NN inference as a sum of a small number (here, 2) of floating point values scaled by very low precision signed integers that represent each individual scalar value in the weights. This scheme goes by the name of BSFP and provides what appears to be an effective means of quantization of weights.

In order to reduce computation overheads in hardware with this additive quantization representation, it is also necessary to represent activations in such a manner so as to reduce the number of full floating-point additions and multiplications. They chose the pre-existing MSFP representation (a chunk of signed integers plus a shared scaling exponent) in order to represent activations. Together, the two representations work with each other to drastically reduce the number of floating-point type operations (significand adders / shifters for multiplication, etc) which provides a direct advantage to hardware PPA. This basic building block to represent an inner product between BSFP and MSFP (called the S2PE) is combined in a systolic array for generalized inner product evaluation.

The work requires some number of parameters to be fixed for HW purposes (e.g., quantization scheme and loss function, precision of the two BSFP components, chunk size, etc), but the scheme appears to provide even more significant compression of the weights over MSFP and improved accuracy on a variety of networks (ShuffleNet-v2, MobileNet-v2, VIT, etc).


**Summary Of The Review:**

Overall, I am borderline accept to full accept, this does add something new to the literature (first time I've seen additive quantization used in this space for one) and as an existence proof with concrete results it is super interesting, but the BSFP technique itself (including the discussion about using signed integers instead of wasted +/-0 representation space with sign/significand for MSFP) do have antecedents. A little more evidence on broader applicability I think would be useful in order for HW people to consider it in their inference designs, but I think that will likely have to wait for others (or the authors themselves) to expand the work in subsequent papers.

---

> ### Author Response · Authors · 2022-11-19
> **Response to Reviewer KfxR**
>
> We sincerely thank you for the insightful questions and comments. Please let us address the concerns and suggestions below.
>
> **[Question 1] In "Hardware Evaluations", it seems that synthesis was run on a 16-wide S2PE, but what exactly was the synthesized design for a 16-wide bf16, int4, int8 design? Does the 16-wide int8 inner product tree produce a 32-bit output or a 8 * 2 + log2(16) = 20 bit output? Where did the bf16 blocks come from (DesignWare or elsewhere)? Any information on the slack in the various designs were produced? How was power estimated, using average switching activity or by using realistic test vector SAIFs from real networks?**
>
> The 16-wide int8 inner product tree produces 20-bit partial sums, and the 16-wide int4 inner product tree produces 12-bit partial sums, and their partial sums are accumulated using a 32-bit accumulator.  We implement BF16 by ourselves, and BF16's internal integer multipliers and adders are from DesignWare.  We estimate the power by synthesizing the RTL designs at the TSMC 40 nm node and using average switching activity. We have listed this information in Appendix F.
>
>
> **[Question 2] Table 1 is a little confusing to look at. The bolded numbers are for all top-1 results of BSFP, though some of the bolded numbers are under the neighboring MSFP numbers. However I would agree that the two techniques are not directly comparable (as BSFP does offer big HW efficiencies). The Pareto frontier charts are simply easier to understand.**
>
> Sorry for the confusion. We have updated Table 1 to use bolded numbers specifically for better metrics, e.g., higher accuracy.  Pareto frontier charts are plotted in Figure 8.
>
> **[Question 3] Regarding "FixOPs (8b)", how is this computed for BSFP as many of the adders / multipliers in the S2PE are sub-8b?**
>
> We normalize sub-8b ops to 8b ones. For example, a 4 by 6-bit multiplication is equivalent to (4*6)/(8*8) = 3/8 FixOP.

---

### Official Review · Reviewer_guHP · 2022-10-29

**Confidence:** 4
**Correctness:** 3
**Technical Novelty And Significance:** 2
**Empirical Novelty And Significance:** 2
**Recommendation:** 5

**Clarity, Quality, Novelty And Reproducibility:**

The proposed method seems to be a new approach to reducing bit-precision for post-training quantization. However, the hardware advantages claimed by the authors are hard to verify, and there is no careful comparison with other hardware implementation scenarios, such as other precision scalable MAC architectures (divide & conquer, sub-word parallel, etc.). In particular, bit-serial architecture is known to be slow due to its multi-cycle arithmetic computation. How does it affect the overall performance?


**Strength And Weaknesses:**

(Strengths)
- A new data format (potentially) more representative of asymmetric distributions.

- Evaluation of quantization accuracy on various ImageNet classification tasks.

(Weaknesses)
- The proposed methods incur more arithmetic operations (e.g., multiplication per subword and addition of psums from the sub-words). The authors claim that these additional computations can be efficiently handled by custom hardware. However, it is very difficult to properly evaluate their claims on hardware benefits without enough evidence and fair comparisons.

- The authors claim several algorithmic advantages of the proposed format, such as adaptation to skewed/non-uniform distributions. However, these claims are not theoretically or analytically supported in the paper; E.g., there is no guarantee that asymmetric number representations of two's complement format are helpful for capturing asymmetric weight distributions. Note that the two's complement is significantly skewed to the negative regions. This fixed skew might be harmful if the weights are positively skewed or not skewed much. There is no analysis in the paper that provides theoretical reasonings about it.




**Summary Of The Paper:**

This paper proposed a new number representation format for efficient post-training quantization of deep neural networks. The proposed format consists of a vector of words each of which is linearly decomposed by two subwords with reduced precision floating point scales. The authors claim that the proposed format is more suitable for representing asymmetric weight distribution. The authors evaluate the proposed method with several ImageNet classification models, and they also presented the hardware implementation for it.

**Summary Of The Review:**

This paper proposed an interesting reduced-precision data representation, but the proposed method incurs hardware overhead. The authors tried to justify their choices with their hardware implementation, but the corresponding discussions are not thoroughly evaluating all the hardware options. Unfortunately, there is little theoretical analysis that justifies the algorithmic advantages of the proposed method, which the readers of ICLR would most appreciate. In this regard, it seems to the reviewer that this paper might fit better for the hardware architecture conferences such as DAC and ISCA.

---

> ### Author Response · Authors · 2022-11-19
> **Response to Reviewer guHP**
>
> Thank you for the valuable review comments. We address your concerns and suggestions below.
>
> **[Weakness 1] The proposed methods incur more arithmetic operations (e.g., multiplication per subword and addition of psums from the sub-words). The authors claim that these additional computations can be efficiently handled by custom hardware. However, it is very difficult to properly evaluate their claims on hardware benefits without enough evidence and fair comparisons.**
>
> Let us take ShuffleNet (Table 1), for example. MSFP requires 7-bit weights and 7-bit activations to achieve 68.27% accuracy, and in comparison, BSFP employs 3+3-bit weights and 6-bit activations and achieves 68.62%.  Although BSFP involves two times more multiplications than MSFP, BSFP's multipliers are 3 by 6-bit in and 9-bit out, but MSFP's are 7 by 7-bit in and 14-bit out.  Qualitatively, it is not unreasonable that the former can be more efficiently realized.
>
> As for hardware comparisons, please refer to the answer to the following question.
>
> **[Clarity 1] There is no careful comparison with other hardware implementation scenarios, such as other precision scalable MAC architectures (divide & conquer, sub-word parallel, etc.). In particular, bit-serial architecture is known to be slow due to its multi-cycle arithmetic computation. How does it affect the overall performance?**
>
> BSFP is not limited to bit-serial architectures. A conventional bit-parallel design can employ 16+16 multipliers to parallelly perform two inner products on BSFP's two vectors of subword mantissas (Fig. 11 in Appendix E).
>
> Advocating bit-serial architecture is not the main point of this work, but bit-serial architecture is indeed a representative vehicle for us to compare BSFP and MSFP.
> Recently, bit-serial AI accelerators have received much attention from premier architecture conferences, just to name a few, Laconic (ISCA 19) and FuseKNA (HPCA 21). Although bit-serial architecture incurs multi-cycle computation, it offers high computing parallelism due to simplified multipliers and facilitates mixed-precision computation.
>
> We have added Appendix F to describe our hardware evaluation methodologies and setups, which are comparable to the related architecture papers, such as Laconic and FuseKNA. Specifically, we realize MSFP, Power-of-Two, BF16, INT4, and INT8-based PEs in Verilog, synthesize the PEs using Synopsys Design Compiler with the TSMC 40 nm standard cell libraries at 500 MHz, and report per-PE area, per-PE power, and iso-area throughput according to the synthesis-stage results.  For example, Figure 9 (left) compares accuracy and iso-area throughput, which is a fair comparison.  Low-precision hardware hurts the accuracy but favors the iso-area throughput. BSFP's tradeoff curve is close to the upper-right corner the most and outperforms the other formats, such as MSFP.
>
>
> **[Weakness 2] The authors claim several algorithmic advantages of the proposed format, such as adaptation to skewed/non-uniform distributions. However, these claims are not theoretically or analytically supported in the paper; E.g., there is no guarantee that asymmetric number representations of two's complement format are helpful for capturing asymmetric weight distributions. Note that the two's complement is significantly skewed to the negative regions. This fixed skew might be harmful if the weights are positively skewed or not skewed much.**
>
> We explain and answer your questions as follows. First, BSFP employs signed scaling factors, so it can flip two's complement to cover positively skewed weights, too. Second, since BSFP employs two sets of two's complement and two scaling factors, weights that are not skewed much can be covered when one scaling factor is set positive and the other is negative. Third, by properly setting the two scaling factors, BSFP can degenerate to uniform and symmetric quantization levels similar to MSFP, and in this sense, BSFP subsumes MSFP.
>
> In addition to the above qualitative explanations, we also did experiments based on real neural networks and weights to quantitatively verify BSFP's advantages (Figure 6).  The results show that BSFP achieves much lower KL divergence than MSFP, which means that BSFP fits the full-precision weights better than MSFP.

---

### Official Review · Reviewer_A7Vh · 2022-11-01

**Confidence:** 5
**Correctness:** 3
**Technical Novelty And Significance:** 4
**Empirical Novelty And Significance:** 2
**Recommendation:** 6

**Clarity, Quality, Novelty And Reproducibility:**

The paper is fairly novel, the proposed quantization format is inspired by multi-level quantization and BFP, but it is a major departure from existing work.

The paper is generally clear, but the process of converting from float to BSFP is not sufficiently detailed. The paper can and should be fixed to include such details so a reader can reproduce the model size and accuracy results.

The hardware throughput results are somewhat unconvincing since it using a hardware architecture that is clearly favoring BSFP. There is also not enough detail to reproduce the hardware results (although there is no way to fix this, we're at ICLR not a architecture conference) I think the paper stands well on the other results and the authors should scale back the hardware claims to better fit the venue/readers.

**Strength And Weaknesses:**

Strength:
 - BSFP is a novel data type which combines multi-level quantization with BFP.
 - BSFP is clearly algorithmically better than BFP since it uses float scale factors. Despite this BSFP does not require float multiplies during computations when multiplied with BFP.
 - Paper is well written. The results surrounding accuracy and model size is convincing. The comparison is made to a strong quantization baseline in MSFP.

Weaknesses:
 - The process of converting floating-point weights to BSFP is not fully clear. The exhaustive algorithm to find the scale factors makes sense, but how to compute both sub-word mantissas? The paper mentions "iterative rounding" but this needs to be made clearer. If the two mantissas are of uneven widths (as is the case in many of the experiments) which mantissa is computed first?
 - Most of the BSFP configurations tested in Table 1 use uneven subword mantissa widths. These configurations may not be efficient outside of bit-serial architectures. This is unlike BFP which is relatively easy to implement even on CPU and GPU.
 - BSFP trades away flexibility for efficiency. Vanilla BFP can be used easily for training and fine-tuning, but BSFP due to the complex quantization procedure can only be used for inference.
 - The throughput comparison between BSFP and MSFP was made on a custom bit-serial processor, which seems like it was designed specifically for BSFP. For example, MSFP uses a wider shared exponent which incurs a greater shift overhead in their design. But I think this is specific to bit-serial and would not be an issue for a conventional adder tree. This makes the throughput comparison somewhat unconvincing. I honestly think the authors should scale back the amount of the paper devoted to the hardware throughput comparison. ICLR is not the venue for this. Focus the paper on the model size and accuracy results instead.

**Summary Of The Paper:**

The authors Block and Subword-Scaling Floating-Point (BSFP), a variant of block-floating point (BFP). BFP represents a vector of real numbers using a single shared exponent and a per-element signed mantissa. BSFP is a variant with the following key differences:
 1. BSFP represents a vector of real numbers as the sum of *multiple* such BFP vectors. The paper only experiments with 2 vectors.
 2. BSFP uses shared low-bit float scaling factors (which can be an arbitrary number) as opposed to a shared exponent (which must be a power of 2)

BSFP is designed for post-training inference. BSFP is only used for the weights, while activations are left in BFP. This is because the weights are known ahead of time, so the authors can use a brute-force exhaustive algorithm to finding the best scaling factors. The dot products between weight and activation will be between BSFP and BFP, which is actually a positive as such a dot product is efficient and does not require floating-point multiplies:
 1. Individual mantissas can be multiplied in fixed-point
 2. The float BSFP scale factor(s) and the power-of-two BFP scale factor can be multiplied by an exponent summation.
 3. The scale factor product can be multiplied onto the dot product sum with a fixed-point mantissa multiply.

The authors propose to use a custom bit-serial architecture to support BSFP. There's a architectural diagram given but not much detail in the text. The architecture is based on a bit-serial processing engine from another paper. Using bit-serial means the subwords in BSFP can be different bitwidths.

The authors compare against MSFP, Microsoft's BFP implementation which seems like a fairly strong baseline. This is a great improvement from many quantization papers which compare only against FP32. Experiments are done over a suite of CNNs and VIT on ImageNet. Convincing results show that BSFP achieves better accuracy-per-model-size compared to MSFP. However, the throughput comparison is done with both MSFP and BSFP on the custom bit-serial architecture - this feels weak to me since bit-serial is not widely used in practice and the MSFP paper uses a conventional accelerator. Still the experiment section as a whole is quite convincing.

**Summary Of The Review:**

The paper proposes BSFP, a novel data format similar to block-floating point (BFP). BSFP uses multiple BFP vectors to represent a single vector of real numbers. BSFP uses floating point scale factors but can efficient multiple with a BFP vector with only fixed-point operations. BSFP has a complex quantization procedure and is only suitable for pre-quantized weights during inference. BSFP was designed for an unconventional bit-serial architecture and may not be efficient on commercial devices. Nevertheless, BSFP is quite novel and achieves improved accuracy per model size when benchmarked against a strong BFP baseline. This makes it an interesting paper for the quantization community.

The biggest change I advise is to focus the paper on the quantization algorithm and data format and away from the hardware architecture.

---

> ### Author Response · Authors · 2022-11-19
> **Response to Reviewer A7Vh**
>
> Thank you for the positive review comments. Please let us address your concerns and suggestions below.
>
> **[Suggestion] The paper can be fixed to include more details on converting from float to BSFP. (i.e., How to compute both sub-word mantissas? The paper mentions "iterative rounding" but this needs to be made clearer. If the two mantissas are of uneven widths, which mantissa is computed first?)**
>
> We have added Appendix D to incorporate the details and pseudo code of the quantization flow. Specifically, they are the following steps.
>
> 1. Enumerate the two scaling factors.
> 2. List the quantization levels the two scaling factors can generate. The number of quantization levels is modest. For example, there are only 64 levels if the two subwords are 2+4 bits.
> 3. Calculate the MSE between the 16 original weights and their nearest quantization levels.  Finding the nearest quantization level in a list is viable as other quantization schemes, such as APoT (ICLR 20), also employ it.
> 4. Keep the scaling factors that achieve the lowest MSE and go to step 1.
>
> **[Weakness 1] Most of the BSFP configurations tested in Table 1 use uneven subword mantissa widths. These configurations may not be efficient outside of bit-serial architectures. This is unlike BFP which is relatively easy to implement even on CPU and GPU.**
>
> First, BSFP is not limited to bit-serial architectures. A conventional bit-parallel design can employ 16+16 multipliers to parallelly perform two inner products on BSFP's two vectors of subword mantissas (Fig. 11 in Appendix E).
>
> Second, BSFP is especially suitable for ASIC accelerators, which represent a significant portion of AI devices nowadays.
>
> Third, newly designed CPUs and GPUs can support BSFP using extended instructions (and the abovementioned bit-parallel circuits). For example, RISC-V CPUs offer official vector extensions and vendor-specific AI extensions (e.g., SiFive's Intelligence Extensions). Such extensions can further incorporate inner product instructions between a BSFP weight vector and an MSFP activation vector.
>
> **[Weakness 2] BSFP trades away flexibility for efficiency. Vanilla BFP can be used easily for training and fine-tuning, but BSFP due to the complex quantization procedure can only be used for inference.**
>
> Thank you for the inspiring question.  Compared to Vanilla BFP, BSFP lowers the flexibility converting from full-precision weights, but BSFP gains richer flexibility to approximate and convert back to full-precision weights. In view of the different characteristics of weights and activations, it should be a reasonable option to adopt BSFP for weights and MSFP for activations.
>
> BSFP focuses on fine-tuning-free, post-training quantization (as well as inference), which are our heard realistic demands from the industry.  Though BSFP-based training is beyond the scope, here we provide a proposal.
>
> First, each BSFP vector is a weighted sum of two Vanilla BFP vectors. Therefore, by letting the scaling factors be trainable parameters, it is possible to leverage a Vanilla BFP framework to train BSFP networks in the first place without resorting to decomposition-based quantization.
>
> Second, one can employ Vanilla BFP training for the majority of training epochs in the beginning and switch to BSFP training only at the last few epochs.
>
> **[Weakness 3] The throughput comparison is somewhat unconvincing. I honestly think the authors should scale back the amount of the paper devoted to the hardware throughput comparison. Focus the paper on the model size and accuracy results instead.**
>
> Thank you for the suggestions.  We will focus the paper more on the model size and accuracy results in the camera-ready version.
>
> In addition, we have added Appendix F to describe our hardware evaluation methodologies and setups, which are comparable to the related architecture papers, such as Laconic (ISCA 19) and FuseKNA (HPCA 21). Specifically, we realize MSFP, Power-of-Two, BF16, INT4, and INT8-based PEs in Verilog, synthesize the PEs using Synopsys Design Compiler with the TSMC 40 nm standard cell libraries at 500 MHz, and report per-PE area, per-PE power, and iso-area throughput according to the synthesis-stage results.  For example, Figure 9 (left) compares accuracy and iso-area throughput, which is a fair comparison.  Low-precision hardware hurts the accuracy but favors the iso-area throughput. BSFP's tradeoff curve is close to the upper-right corner the most and outperforms the other formats, such as MSFP.

---

> > ### Comment · Reviewer_A7Vh · 2022-11-28
> > **Maintaining score**
> >
> > I thank the authors for their response. I do think the paper has merit, but I maintain that BSFP will be difficult/inefficient in conventional devices like CPUs and GPUs. Yes, you can fit two BSFP narrow multiplications into a single 16x16 multiplier, but this is not an efficient use of such a multiplier. Furthermore, newer Nvidia GPUs use tensor cores which compute matmuls instead of individual multipliers. BSFP would require very specific hardware, and such hardware would not efficiently compute non-BSFP/MSFP data types.
> >
> > I think the paper is a borderline accept.

---

> > > ### Author Response · Authors · 2022-12-08
> > > **Additional Response to Reviewer A7Vh**
> > >
> > > We thank the reviewer for the additional feedback.  We agree with the reviewer that BSFP will be difficult/inefficient in existing devices including existing CPUs and GPUs, but we guess the reviewer will also agree that BSFP can be efficiently deployed in newly-manufactured devices, including customized AI accelerator IPs, which represent a significant amount of AI products including smartphones and smart appliances. Our explanations are as follows.
> > >
> > > One main application field of BSFP is customized, embedded AI accelerator IPs, e.g., Nvidia NVDLA, Qualcomm Hexagon Tensor Accelerator, and MIT Eyeriss. Such IPs are embedded in emerging AI chips for smartphones, appliances, robots, etc.  Since customized, embedded AI accelerator IPs are not obligated to support standard data types, the proposed BSFP type, which improves weight quantization quality (including post-training quantization, which is even difficult and desirable), will stand out as a candidate for chip designers to consider and adopt in their new design.
> > >
> > > As for CPUs, although they are not our only focus, letting newly-manufactured CPUs support BSFP is not difficult, and the computation will be efficient.  Modern CPUs, including RISC-V, ARM, and Synopsys ARC, allow custom ISA extensions [R1,R2,R3].  Designers can provide an add-on digital circuit that natively and efficiently supports BSFP using low-bit, fixed-point adders and multipliers.  Vendors already offered the required design/simulation flow, CPU generator, and toolchain generator to facilitate designers to employ the extended instructions, e.g., vector product instructions between BSFP weights and MSFP activations.
> > >
> > > Thank you for raising the questions, and we will clarify these points in the final manuscript.
> > >
> > > [R1] https://riscv.org/wp-content/uploads/2019/02/Imperas-EW-2019-Custom-Instructions-booth-slides-KM.pdf
> > >
> > > [R2] https://developer.arm.com/Architectures/Arm%20Custom%20Instructions
> > >
> > > [R3] ARC Processor EXtension (APEX) technology

---

### Author Response · Authors · 2022-11-19
**General response to the reviewers**

We thank all the reviewers for the detailed reviews and valuable feedback.

We are glad that you recognized the novelty in our proposed BSFP, the overall clarity of the paper, and the robustness and significance of our results.

The main concern identified in the reviews was the lack of clarity in the quantization flow, the search process for BSFP configurations, and the hardware evaluation setups. As a result, we have made several modifications in these respects, such as adding pseudo code and detailed descriptions in the Appendices.
Some additional issues were raised, which we have addressed in the individual answers. Every modification to the manuscript is highlighted in blue.

Should there be any additional comments, we are available to address them before the end of the discussion period.

Thank you once again,

The authors

---

### Decision · Program_Chairs · 2023-01-20

**Decision:**

Accept: poster

**Justification For Why Not Higher Score:**

This is a borderline accept paper, which is interesting to the hardware research part of the ICLR community, which I imagine is not too big.

**Justification For Why Not Lower Score:**

The paper has merit and contributes to hardware and number representation research for (more efficient) machine learning. There are flaws but there are minors or fixable in subsequent research.

**Metareview: Summary, Strengths And Weaknesses:**

The paper presents a new floating point format: Block and Subword-Scaling Floating-Point (BFSP), which is an extension to BFP (existing). A strong result of the paper is that BFSP outperforms MSFP (a correct, strong baseline to compare to) on ImageNet classification "by up to 20.56% top-1 accuracy at the same weight precision and reduces up to 10.3% model size". A result that is more dependent on the author's hardware implementation choices (which is thoroughly discussed by the reviewers): "BSFP outperforms MSFP by up to 2.0 computing throughput and up to 5.3 energy efficiency under the same silicon area budget".

Strengths:
* clear paper
* BFSP is useful, potent,
  * the claims about its bits effectiveness are supported by evidence,
  * it is compatible with BFP and seems generally flexible/applicable (for a specialist FP format).

Weaknesses:
* [small experimental weakness] the claims about energy efficiency are reasonable but also hardware implementation dependent. This is a somewhat expected of a specialized format.
* [limitation] BFSP is applicable only to trained fixed weights, not necessarily to activations (for which the authors use BFP, which is very compatible). => this limits its applicability to inference only, as recomputing the scaling factors cannot currently be done "online".

What might be missing: more details on the hardware implementation (but ICLR may not the best venue for that), more evidence of applicability.

**Note From Pc:**

if the above contains the word "oral" or "spotlight" please see: "oral" presentation means -> notable-top-5% and "spotlight" means -> notable-top-25%. As stated in our emails, we are disassociating presentation type from AC recommendations

**Summary Of Ac-Reviewer Meeting:**

[We couldn't find a time slot.]